# An Integrated Strategy for the Prevention of SARS-CoV-2 Infection in Healthcare Workers: A Prospective Observational Study

**DOI:** 10.3390/ijerph17165785

**Published:** 2020-08-10

**Authors:** Anna Maria Cattelan, Lolita Sasset, Eugenia Di Meco, Silvia Cocchio, Francesco Barbaro, Silvia Cavinato, Samuele Gardin, Giovanni Carretta, Daniele Donato, Andrea Crisanti, Marco Trevenzoli, Vincenzo Baldo

**Affiliations:** 1Infectious Diseases Unit, Department of Medicine, Azienda Ospedale Università di Padova, 35128 Padova, Italy; lolita.sasset@aopd.veneto.it (L.S.); edm1984@gmail.com (E.D.M.); francesco.barbaro@aopd.veneto.it (F.B.); silvia.cavinato@aopd.veneto.it (S.C.); samuele.gardin92@gmail.com (S.G.); marco.trevenzoli@aopd.veneto.it (M.T.); 2Department of Cardiac Thoracic Vascular Sciences and Public Health, University of Padua, 35128 Padova, Italy; silvia.cocchio@unipd.it (S.C.); vincenzo.baldo@unipd.it (V.B.); 3Health Department, Azienda Ospedale Università di Padova, 35128 Padova, Italy; giovanni.carretta@aopd.veneto.it (G.C.); daniele.donato@aopd.veneto.it (D.D.); 4Clinical Microbiology and Virology Unit, Department of Molecular Medicine, Azienda Ospedale Università di Padova, 35128 Padova, Italy; andrea.crisanti@unipd.it; 5Department of Life Science, Imperial College London, London SW7 2AZ, UK

**Keywords:** SARS-CoV-2, novel coronavirus, infection control, health care workers, personal protective equipment, respiratory protective device

## Abstract

Background: Since the beginning of SARS-CoV-2 outbreak, a large number of infections have been reported among healthcare workers (HCWs). The aim of this study was to investigate the occurrence of SARS-CoV-2 infection among HCWs involved in the first management of infected patients and to describe the measures adopted to prevent the transmission in the hospital. Methods: This prospective observational study was conducted between February 21 and April 16, 2020, in the Padua University Hospital (north-east Italy). The infection control policy adopted consisted of the following: the creation of the “Advanced Triage” area for the evaluation of SARS-CoV-2 cases, and the implementation of an integrated infection control surveillance system directed to all the healthcare personnel involved in the Advance Triage area. HCWs were regularly tested with nasopharyngeal swabs for SARS-CoV-2; body temperature and suggestive symptoms were evaluated at each duty. Demographic and clinical data of both patients and HCWs were collected and analyzed; HCWs’ personal protective equipment (PPE) consumption was also recorded. The efficiency of the control strategy among HCWs was evaluated identifying symptomatic infection (primary endpoint) and asymptomatic infection (secondary endpoint) with confirmed detection of SARS-CoV-2. Results: 7595 patients were evaluated in the Advanced Triage area: 5.2% resulted positive and 72.4% was symptomatic. The HCW team was composed of 60 members. A total of 361 nasopharyngeal swabs were performed on HCWs. All the swabs resulted negative and none of the HCWs reached the primary or the secondary endpoint. Conclusions: An integrated hospital infection control strategy, consisting of dedicated areas for infected patients, strict measures for PPE use and mass surveillance, is successful to prevent infection among HCWs.

## 1. Introduction

Since December 2019, a novel coronavirus (named SARS-CoV-2) drew the attention of the World Health Organization (WHO), and the disease caused by the virus (named COVID-19) started to raise concern and to progressively spread all around the world. From the city of Wuhan (China), the virus has subsequently reached over 110 countries, including the European region [1]. In Italy, the first detection of local transmission of the virus occurred on February 21. A large outbreak rapidly hit northern Italy, with confirmed transmissions in several municipalities [2]. In the Veneto region, the first case and the first Italian death linked to COVID-19 was detected in the village of Vò, located near Padua, on February 22 [3]. As of May 6, 2020, a total of 18,479 cases have been reported in the whole Region and 3879 in the Province of Padua [4].

Since the beginning of the outbreak, a substantial number of COVID-19 cases has been reported among healthcare workers (HCWs) [5,6,7,8,9,10]. The consequences of this are relevant not only for HCWs and their families’ health, but also for healthcare service efficiency maintenance and for the risk of in-hospital transmission of the disease [11]. In Italy, more than 24,358 HCWs have tested positive, and, according to professional associations, more than 150 Italian physicians and 40 nurses have died because of COVID-19 [12]. These data confirm the high risk of healthcare professionals and the compelling need for the implementation of a wider strategy for infection control in healthcare settings. The strategy should include adequate supplies and strict protocols for the use of personal protective equipment (PPE) [13], a rational re-elaboration of patients’ flow and comprehensive measures to ensure adequate working conditions and enlarged screening program for HCWs [14].

The aim of the study was to investigate the occurrence of SARS-Cov-2 transmission among HCWs involved in the first management of suspected or confirmed COVID-19 patients in Padua University Hospital (located in north-east Italy) in order to describe the evidence-based practices adopted to prevent the transmission and to analyze their effectiveness.

## 2. Materials and Methods

This is a prospective observational study between February 21 and April 16, 2020 (week 8–16 of 2020) in Padua University Hospital. During the study period, we collected and analyzed the activities carried out in the “Advanced Triage of the Infectious Diseases Unit” and the results of the infection control-integrated surveillance system implemented to prevent and monitor infection transmission among HCWs. All participants were informed of the purpose and procedures of the study and gave verbal informed consent. The study was conducted in accordance with the Declaration of Helsinki and Ethical approval for this study was obtained from the Institutional Review Board of the University of Padua. Participation was voluntary; subjects could withdraw at any time and all analyses were carried out on anonymized data.

### 2.1. Setting and Operating Model

Padua University Hospital, located in Veneto Region (North-East Italy), offers a tertiary service and is an infectious diseases reference clinical and laboratory center in the regional network. In response to the emerging epidemic in China, at the beginning of February, the hospital prepared an emergency plan that was quickly implemented by the end of the month.

An integrated infection control policy was adopted. It consisted of the following:The predisposition of a fast triage prior to entering the hospital;The creation of separated and dedicated areas to avoid the interaction between potentially infected and non-infected patients;The predisposition of multiple installations for hand disinfection;The application of strict requirements for PPE usage and the implementation of training protocols directed to HCWs;The implementation of an integrated surveillance system to prevent and monitor infection transmission among HCWs.

The newly installed area was called the “Advanced Triage of the Infectious Diseases Unit”. It is a field hospital composed of 10 tents located outside the Infectious Diseases Unit (Figure 1).

The Advanced Triage was created for the evaluation of suspected SARS-CoV-2 cases and contacts, and the initial management of confirmed patients. It was open 24 h a day and 7 days a week. Any patient presenting at the Advanced Triage Unit was invited to disinfect the hands and to a wear a surgical mask and gloves. A clinical assessment including the measurement of body temperature and oxygen saturation was then performed. A patient presenting one of more of the following symptoms was considered “symptomatic”: fever; cough; brachypnea, tachypnea or other signs of respiratory distress; headache; hemoptysis; and diarrhea. According to a physician’s judgment, symptomatic patients were also invited to perform a chest X-ray in a dedicated tent. A nasopharyngeal swab for SARS-CoV-2 was obtained for all the subjects.

After the evaluation, asymptomatic subjects were discharged and put in home quarantine until the result of the swab. Symptomatic subjects were moved to another tent via a transition zone while waiting for the nasopharyngeal swab result. Once a diagnosis of COVID-19 was confirmed, the patients were immediately transferred to the Infectious Diseases ward. In the case of deterioration of the clinical conditions, the patients were moved to the Intensive Care Unit, or other appropriate wards, pending the test results. The area was sanitized with chlorine-based disinfectants after the transition of any positive patient.

Strict requirements for the use of PPE were applied inside and around the Advanced Triage area. The PPE provided for every working included:Filtering face-piece respirators class 2 or 3 (FFP2–3);Personal reusable face shields for HCWs performing nasopharyngeal swabs or visiting symptomatic subjects;Gloves;Water-resistant long-sleeved gowns.

Surgical masks were used only in the case of shortage of FFP masks after a case-by-case assessment. Disposable PPE was changed after the evaluation of serious or critical patients and when invasive procedures were needed (e.g., intubation, central lines insertion, etc.). Face shields were reusable and disinfected according to the instructions of the manufacturer.

### 2.2. Infection Control Surveillance System for HCWs

All HCWs employed in the “Advanced Triage of the Infectious Diseases Unit” were invited to participate in the study and the unique inclusion criteria was to provide informed consent.

HCWs providing first care to COVID-19 patients were proactively monitored: body temperature was measured twice daily, and suggestive symptoms were evaluated at the beginning of each duty. In the case of fever and/or respiratory symptoms, the staff members were invited to stay at home. If they developed symptoms during their duty, they were promptly isolated and tested. In addition, even in the absence of symptoms, all the workers were tested with nasopharyngeal swabs for SARS-CoV-2 once every 7–10 days.

All HCWs received a mandatory SARS-CoV-2 prevention training, which covered also the correct use of PPE. Each staff member was required to produce a documented evidence of having participating in the training.

All HCWs’ data were collected in clinical records within the proactive infection control tool already used in our hospital for healthcare personnel.

To evaluate the results and the effectiveness of the surveillance system, the following endpoints were defined:Primary endpoint: symptomatic infection (presence of one or more of the following symptoms: fever; cough; brachypnea, tachypnea or other signs of respiratory distress; headache; hemoptysis; and diarrhea) with confirmed detection of SARS-CoV-2 at the nasopharyngeal swab;Secondary endpoint: asymptomatic infection with confirmed detection of SARS-CoV-2 at the nasopharyngeal swab.

The registry of sick requests was consulted and, in the case of absence from work of the HCWs, a phone call was made to evaluate the need to carry out the swab in relation to the symptoms as defined by a regional protocol.

### 2.3. Nasopharyngeal Swabs

Nasopharyngeal swabs were performed by using flocked swabs in a liquid-based collection and transport systems (eSwab^®^, Copan Italia Spa, Brescia, Italy). All nasopharyngeal swab samples were processed with an in-house Real-Time Polymerase Chain Reaction (RT-PCR) method according to Lavezzo et al. [15]. All tests were performed at the Clinical Microbiology and Virology Unit of Padua University Hospital, which is the regional reference laboratory for emerging viral infection.

The average time from sampling to response was calculated.

### 2.4. Statistical Analysis

Data were presented as percentages for categorical variables and mean ± SD (standard deviation) for continuous variables, which were compared using Student’s t test for unpaired data and performing a priori tests for equality of variances. A linear regression model was used to analyze the correlation between the number of positive nasopharyngeal swabs and the number of tests performed on HCWs. A *p* value < 0.05 was accepted as statistically significant. Statistical analysis was performed using the SPSS version 25 software for Windows (International Business Machines Corporation, Armonk, NY, USA).

## 3. Results

During the 8 weeks, a total of 7595 subjects were examined and tested in the Advanced Triage. A total of 395 (5.2%) patients showed positive results, and 52.5% of the total sample of patients was male. The mean age of positive subjects was 53.9 ± 18.2 years, compared to 47.6 ± 15.9 of negative subjects (*p* < 0.001). The number of subjects tested for SARS-CoV-2 and the percentage of positive patients attending the Advance Triage Unit according to age group is as shown in Figure 2. An amount of 286 of the 395 positive subjects (72.4%) were symptomatic and admitted to the Infectious Diseases ward.

Overall, 12,822 tests were performed—approximately 228 swabs/day (276 on weekdays and 108 over the weekend). The peak of the performed tests was reached in week 12. The rate of positivity remained stable at about 6.0% until week 12, and decreased from then to arrive at 1.2% in week 16 (Figure 3).

A total of 544 nasopharyngeal swabs (4.3%) resulted positive for SARS-CoV-2. Overall, the median time from sampling to response in symptomatic subjects was 10.33 h (range: 3.18–24.06).

All HCW teams employed in the Triage Unit were included in the study. The staff was composed of 21 physicians (35%), 29 nurses (48.3%) and 10 nursing support workers (16.7%). The average age of the HCWs was 46.2 ± 11.6 years and 80% were females (48/60). The organization of shift work is shown in Table 1. Overall, the staff worked for 122 h a day shared in 47 h, 40 h and 35 h for physicians, nurses and support workers, respectively.

During the study period, 361 nasopharyngeal swabs were performed on HCWs—on average 6.0 swabs for each member of the staff (Table 2). All the swabs performed on HCWs resulted negative for SARS-CoV-2, and none of the HCWs reported symptoms of infection showing a “zero” SARS-CoV-2 transmission rate among the HCW teams. In addition, there was no increase in sick leaves requests from HCWs for diseases potentially associated with coronavirus infections (e.g., influenza-like illnesses, pneumonia, cardiovascular diseases, etc.).

The number of tests performed each day on HCWs remained stable during the study period and was not influenced by the proportion of SARS-CoV-2-positive tests. The regression analysis did not show a statistically significant association between the rate of positivity and the number of nasopharyngeal swabs performed on HCWs (R^2^ = 0.025; *p* = 0.27) (Figure 4).

Finally, the amount of PPE used during the study period is shown in Table 2.

## 4. Discussion

Since the very beginning of the SARS-CoV-2 outbreak, an integrated infection control policy was adopted at the University Hospital of Padua. It mainly consisted of the creation of a dedicated Advanced Triage area for the first management of patients and of the implementation of a series of measures aimed at the minimization of the risk of infection transmission. In the first eight weeks, almost 7600 subjects were examined and tested in the Advanced Triage area. More than 5% of the subjects tested positive for coronavirus infection and, among them, more than 70% presented symptoms of various degrees. A team of 60 health professionals, including specialized physicians, doctors in training, nurses, and support workers, were employed in the Advanced Triage. All the HCWs were subject to an integrated infection control surveillance system, and none of them reported clinical or virologic evidence of infection.

When epidemics of highly infectious diseases such as SARS-CoV-2 occur, HCWs are at much greater risk of infection than the general population. This is mainly due to their constant and close contact with patients’ contaminated body fluids [9,10]. The tribute of HCWs to the COVID-19 outbreak has indeed been very significant. A total 3.8% of the confirmed COVID-19 cases reported in China were HCWs, of which 14.8% presented severe or critical diseases and 5% died [5]. In European countries between 9% and 26% of all COVID-19 cases are in HCWs [6]. A recent study conducted in the Netherlands found a rate of positivity of 6% for COVID-19 among HCWs [7]. In Italy, the rate of positive HCWs is around 9%, but higher figures (up to 20%) are reported in the Lombardy region [8,9].

Unlike most previous reports, our study has not shown cases of SARS-CoV-2 infection in physicians, nurses or support workers during the observational period. We believe that this was the result of a successful comprehensive strategy implemented since the beginning of the epidemic. The reorganization of patients’ flows, the establishment of new management procedures, the proper use of PPE and the surveillance program for HCWs all played an important role. A recent study performed in China identified the following as risk factors for infection in HCWs: working in high-risk settings (respiratory and infection departments, intensive care units or surgical departments), suboptimal infection control practices (such as hand hygiene) and longer duty hours [10]. A recent literature review also highlighted the role of PPE use in risk reduction [16].

In our setting, SARS-CoV-2 infection risk for HCWs was potentially high because of the high number of patients that visited per day and the high proportion of performed positive nasopharyngeal swabs. Moreover, 72.4% of patients were symptomatic for fever, cough and other respiratory tract symptoms. An additional fact to be considered is that symptomatic patients were requested to wait for the result of the test in the dedicated area for an average of 6–8 h under continued nursing monitoring. Overall, the median time of exposure of HCWs to potential risk was estimated to be around 10 h. The inter-personal distance and the time spent with confirmed infected persons influence the effectiveness of human-to-human transmission of SARS-CoV-2 virus in healthcare settings [17,18]. In our tent structures, an adequate ventilation was guaranteed by open windows, as can be seen in Figure 1. It is also well known that SARS-CoV-2 is transmitted through large respiratory droplets (inhaled or deposited on mucosal surfaces) and that other routes of transmission include contact with contaminated fomites and inhalation of aerosols produced during aerosol-generating procedures (AGPs). With the exception of AGPs, it is unclear whether FFP respirators (class 2 or 3) provide a better protection than surgical masks against SARS-CoV-2 and other respiratory viruses. Quantified protection analyses and studies among healthcare individuals working with patients suspected of respiratory illness are limited [19]. In a randomized clinical trial performed on the effectiveness of FFP2 respirators and surgical masks, Radonovich and colleagues reported no significant difference in the incidence of laboratory-confirmed influenza between HCWs using these two types of equipment [20]. However, it has been recently reported that SARS-CoV-2 can potentially be dispersed in fine, airborne particles, and that viable viruses can persist for 3–4 h in the air, suggesting that respiratory droplets precaution might be not appropriated for SARS-CoV-2 [19]. In our setting the use of FFP2 respirators by HCWs and of surgical masks by all the patients guaranteed an adequate protection against SARS-CoV-2 transmission. As recommended by the European Centre for Disease prevention and Control (ECDC) guidelines, HCWs in contact with suspected or confirmed COVID-19 cases were also required to wear eye protection (i.e., visors or goggles), long-sleeved gowns and gloves [21]. Furthermore, the procedures of putting on (donning) and safely removing (doffing) PPE and hand hygiene practices were actively assisted and controlled by a trained member of the staff. This aspect has been recently described by Xuejiao Chen and colleagues, who showed that the implementation of a proactive infection control tool is able to provide a real-time monitoring and a consequent rapid bad-practice correction [22].

In our study, PPE daily use was slightly higher than the estimated PPE needs reported by the ECDC [21]. However, our Advanced Triage setting was not comparable to a simple isolation area: patients required different levels of care and the grade of inter-personal contact was high due to the frenetic respiratory sample collection activity performed in enclosed spaces.

It has been described that the use of FFP2 respirators is associated with more adverse effects (such as skin reactions, dry mouth and cough) compared to surgical masks [23]. In our experience, despite some discomforts reported by HCWs, the adherence remained high. This was strongly influenced by risk perception and by the long experience of our staff with airborne transmitted infections.

The proactive infection control program implemented for all the HCWs of the Advanced Triage consisted of an integrated syndromic and virologic surveillance system. It reduced unnecessary isolation, contact tracing necessity and anxiety, factors that can be highly challenging during an outbreak situation [24]. We suggest that the high and steady frequency of nasopharyngeal swabs offered during the whole period helped the staff to maintain a good control of both physical and mental health. In fact, the number of tests performed on HCWs remained stable, and was not related neither to the burden of patients visiting the Advanced Triage nor to the weekly rate of confirmed COVID-19 cases. Another possible contributing factor could have been the good planning of working shifts and the avoidance of prolonged duty hours. Notably, we did not observe any reductions in services’ performance (e.g., delays or disruptions), nor an excess morbidity possibly associated with SARS-CoV-2 infections among the staff.

Our study has some limitations. First, it was limited in scope: the HCWs involved in the study where all employed in the Advanced Triage, limiting the generalization of our findings to all the workers of the hospital. Second, the study was carried out over 8 weeks and lacks longitudinal follow-up. Third, an assessment of mental health of HCWs was not performed. Nevertheless, we offer an accurate description of all the measures for the first management of COVID-19 patients in the hospital, and our findings support current recommendations for PPE use and for the need for a continuous and integrated mass surveillance of HCWs.

## 5. Conclusions

Containment of COVID-19 remains a great challenge in both community and healthcare settings. In particular, infection prevention and control practices in hospitals play a critical role in the epidemic chain. HCWs are key subjects in controlling the epidemic, ensuring the function and the operativity of the healthcare services and minimizing the risk of infection spreading in the hospital and in the community. It can be fairly affirmed that a good control of COVID-19 dissemination goes together with the protection of HCWs. This is the reason why hospital boards, stakeholders and health policymakers should make concerted effort to provide HCWs all the support measures needed and to ensure adequate supplies of protective equipment. Those aspects are essential and enable HCWs to safely face the continuous, exhausting and stressful daily work that characterizes an outbreak situation.

## Figures and Tables

**Figure 1 ijerph-17-05785-f001:**
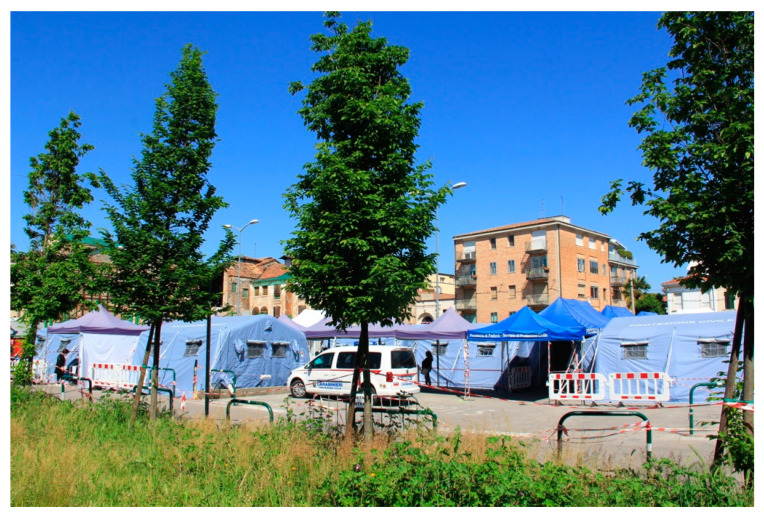
The Advanced Triage of the Infectious Diseases Unit, Padua University Hospital. (Reproduced with permission from Azienda Ospedale Università di Padova).

**Figure 2 ijerph-17-05785-f002:**
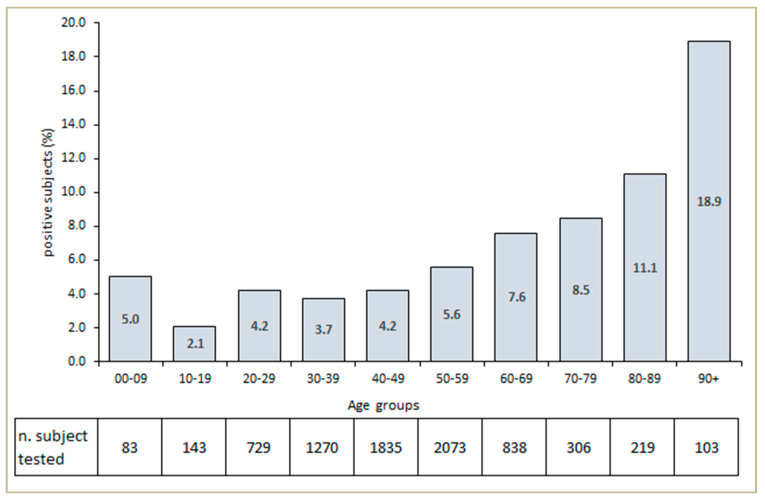
Number of subjects tested and percentage of subjects who tested positive for SARS-CoV-2 and attended the Advanced Triage Unit according to age group.

**Figure 3 ijerph-17-05785-f003:**
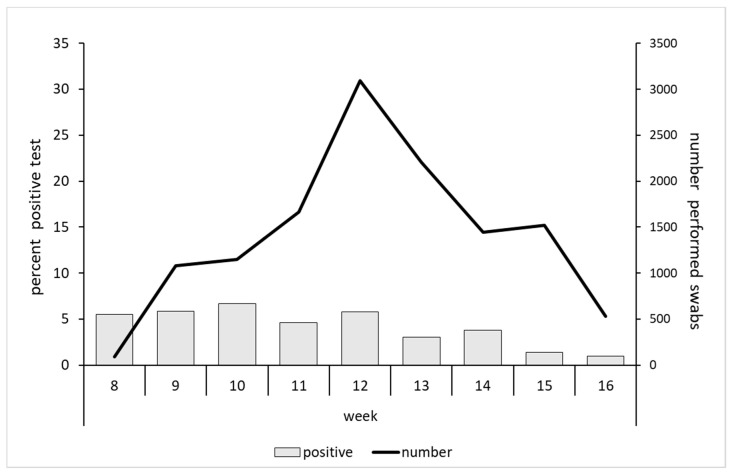
Number of performed nasopharyngeal swabs and percentage of SARS-CoV-2-positive tests according to weeks.

**Figure 4 ijerph-17-05785-f004:**
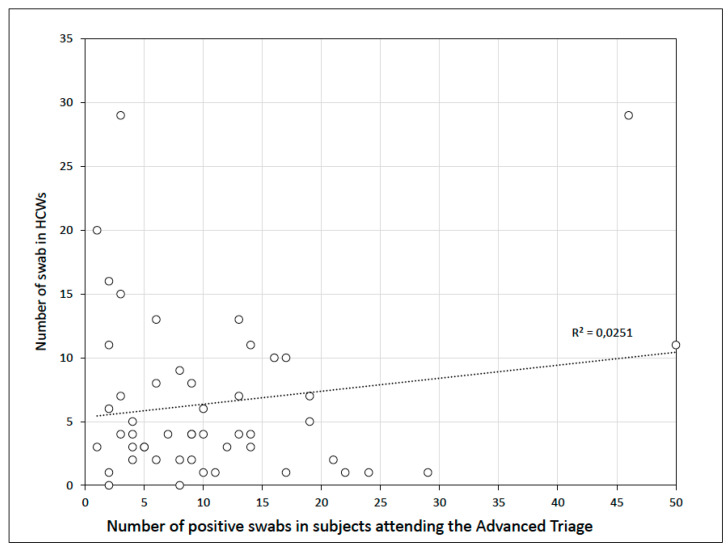
Linear regression analysis between the total positive nasopharyngeal swabs and the number of swabs performed on healthcare workers (HCWs). Each point represents a day of the period study.

**Table 1 ijerph-17-05785-t001:** Day working shifts for healthcare workers of the Advanced Triage of the Infectious Diseases Unit.

z	A.M.												P.M.											
	01:00	02:00	03:00	04:00	05:00	06:00	07:00	08:00	09:00	10:00	11:00	12:00	01:00	02:00	03:00	04:00	05:00	06:00	07:00	08:00	09:00	10:00	11:00	00:00
1 Physician																								
1 Physician																								
1 Physician																								
1 Volunteer physician																								
2 Resident doctors																								
2 Resident doctors																								
3 Nurses (for every 8 h shift)																								
7 Nurses																								
7 Nurses																								
1 Nursing support worker																								
1 Nursing support worker																								
2 Nursing support workers																								
1 Nursing support worker																								

**Table 2 ijerph-17-05785-t002:** Nasopharyngeal swabs performed on the healthcare workers and personal protective equipment used during the 8 weeks study period.

	**Number of HCWs**		**Number of Swabs**	**Average Number of Swabs per Person**
**Physicians**	21		90	4.3
**Nurses**	29		206	7.1
**Support Workers**	10		65	6.5
**Total**	60		361	6.0

**Personal Protective Equipment**		**Number of Items Used**		**Number of Items/Suspected Cases**
Long-sleeved Water-Resistant Gown		1904		0.15
Respiratory Protection (FFP2 or FFP3)		2514		0.20
Gloves		54536		4.3

HCWs: healthcare workers; FFP: filtering face piece mask.

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
