# Peer review of "An Integrated Strategy for the Prevention of SARS-CoV-2 Infection in Healthcare Workers: A Prospective Observational Study"

_ijerph, 2020, doi:10.3390/ijerph17165785_

Round 1

Reviewer 1 Report

Dear authors, after reviewing the manuscript, I submit the following comments.

Best regards,

The Abstract section:

In line 36, it does not specify when, how (evaluating the health status of the HCWs) the research was carried out, what variables were used, etc.

In line 39, “To evaluate the efficiency of the control strategy among HCWs we defined a primary endpoint (symptomatic infection with confirmed detection of SARS-CoV-2) and a secondary endpoint (asymptomatic infection with confirmed detection of 41 SARS-CoV- 2). ”, Is more a possible objective than a description of the research methodology used.

The Material and methods section:

In line 80, they present “All participants gave verbal informed consent and all analyzes were carried out on 80 anonymised data”, which is a serious problem in the investigative methodology, because they do not have a documentary trace of said verbal informed consent (which Information has been given to the subject under investigation, up to which consent is requested –including seeing the registry that records the diseases of the PS-, obligations and rights, including requesting to leave the study) and will handle health data, considering it sensitive data. , within a public health center, that could violate the European data protection directive (Regulation (EU) 2016/679).

The period of February 21 and April 16, 2020, is it a period of complaint, data collection? You must specify both, if they are different.

In section 2.1 setting and operating model, you have correctly described the healthcare protocol and the control measures on the control of the Healthcare Personnel involved, under "An integrated infection control policy was adopted.", But you have not described the protocol for research and follow-up, within the retrospective study, such as where the health data of the HCWs were obtained, were given it or accessed a record in the healthcare unit (proactive infection control tool among PS).

You had described the characteristics that the staff working in these services (training, etc.) must have, but has not indicated the eligibility criteria and the sources and methods of selection of the participants: are they all staff, was recruitment random of the participants, massive or intentional? to assess selection bias. Describe the monitoring methods. The inclusion and exclusion criteria of the PS included in the study ?. I encourage you to describe them.

In the data obtained from the patients, were they total or from those attended by the professionals participating in the study ?, in case the entire PS of the unit did not participate, due to the selection method used. I ask you to clarify

It does not describe the variables used within this section.

You had stated that “Simple t test was used to compare continuous variables”. However, they do not clarify what tests they used to assess the relationship between categorized (eg, week number) and continuous variables. Would they be parametric or non-parametric? If yes, did you previously use the homogeneity or co-variance test?

Although it is on line 217, within the results section, it does not explain in this section, where the data of the casualties were obtained and if they asked for permits to those affected, within and if it was included in the verbal informed consent.

The References section:

It can be seen that the bibliographic references 1, 3, 4, 6, 9, 12, 13, 18-20, are poorly formulated.

Among the most frequent failures are the missing data in the references of web pages consulted, wrongly put the month and day of publication after the year, the name of the magazines abbreviated with points, doi (in all).

For this reason, I encourage you to address the journal's specific reference standards.

Kinds regards

Author Response

Dear Reviewer,

we appreciated your comments and adjusted the manuscript according to your suggestions.  

The Abstract section:

In line 36, it does not specify when, how (evaluating the health status of the HCWs) the research was carried out, what variables were used, etc.

Answer: we have modified the text as follows:

“This prospective observational study was conducted between February 21 and April 16 2020, in the Padua University Hospital (north-east Italy). The infection control policy adopted consisted of: the creation of an Advanced Triage area for the evaluation of SARS-CoV-2 cases; the implementation of an integrated infection control surveillance system directed to all the Healthcare Personnel involved in the Advance Triage area.  HCWs were regularly tested with nasopharyngeal swabs for SARS-CoV-2, body temperature and suggestive symptoms were evaluated at each duty. Demographic and clinical data of both patients and HCWs were collected and analyzed; HCWs personal protective equipment consumption was also recorded”.

In line 39, “To evaluate the efficiency of the control strategy among HCWs we defined a primary endpoint (symptomatic infection with confirmed detection of SARS-CoV-2) and a secondary endpoint (asymptomatic infection with confirmed detection of 41 SARS-CoV- 2). ”, Is more a possible objective than a description of the research methodology used.

Answer: The sentence was changed as:  

“The efficiency of the control strategy among HCWs was evaluated identifying symptomatic infection and asymptomatic infection with confirmed detection of SARS-CoV-2”.

The Material and methods section:

  1. The period of February 21 and April 16, 2020, is it a period of complaint, data collection? You must specify both, if they are different.

Answer: The sentence was integrated as follows:

This is a prospective observational study conducted between February 21 and April 16, 2020 (week 8-16 of 2020) in Padua University Hospital. During the study period we recorded and analyzed the activities carried out in the “Advanced Triage of the Infectious Diseases Unit”.

  1. In line 80, they present “All participants gave verbal informed consent and all analyzes were carried out on anonymised data”, which is a serious problem in the investigative methodology, because they do not have a documentary trace of said verbal informed consent (which Information has been given to the subject under investigation, up to which consent is requested –including seeing the registry that records the diseases of the PS-, obligations and rights, including requesting to leave the study) and will handle health data, considering it sensitive data. , within a public health center, that could violate the European data protection directive (Regulation (EU) 2016/679).

Answer: We have considered the Editorial notes. We would like to underline that HCWs volunteered en masse to this study. In addition, in our hospital there’s a specific Unit dedicated to the control and safety of HCWs and a complete database is available.

We have also integrated the text as follows:

“All participants were informed of the purpose and procedures of the study and gave verbal informed consent. The study was conducted in accordance with the Declaration of Helsinki and Ethical approval for this study was obtained from the Institutional Review Board of the University of Padua. Participation was voluntary; subjects could withdraw at any time and all analyses were carried out on anonymised data”.

  1. In section 2.1 setting and operating model, you have correctly described the healthcare protocol and the control measures on the control of the Healthcare Personnel involved, under "An integrated infection control policy was adopted", But you have not described the protocol for research and follow-up, within the retrospective study, such as where the health data of the HCWs were obtained, were given it or accessed a record in the healthcare unit (proactive infection control tool among PS).”

                                                                                    +

“You had described the characteristics that the staff working in these services (training, etc.) must have, but has not indicated the eligibility criteria and the sources and methods of selection of the participants: are they all staff, was recruitment random of the participants, massive or intentional? to assess selection bias. Describe the monitoring methods. The inclusion and exclusion criteria of the PS included in the study? I encourage you to describe them.”

                                                                                    +

“In the data obtained from the patients, were they total or from those attended by the professionals participating in the study ?, in case the entire PS of the unit did not participate, due to the selection method used. I ask you to clarify. It does not describe the variables used within this section”

Answer: It was a prospective observational study and the follow-up of the HCWs was described in section 2.2 where we introduced the following sentences:

“All HCWs employed in the “Advanced Triage of the Infectious Diseases Unit” were invited to participate in the study and the unique inclusion criteria was to provide informed consent.

“All HCWs data were collected in clinical records within the proactive infection control tool already used in our hospital for Healthcare Personnel.

  1. “You had stated that “Simple t test was used to compare continuous variables”. However, they do not clarify what tests they used to assess the relationship between categorized (eg, week number) and continuous variables. Would they be parametric or non-parametric? If yes, did you previously use the homogeneity or co-variance test?”

Answer: We completed the sentences with:

“for continuous variables, which were compared using Student’s t test for unpaired data, performing a priori tests for equality of variances”.

5.“Although it is on line 217, within the results section, it does not explain in this section, where the data of the casualties were obtained and if they asked for permits to those affected, within and if it was included in the verbal informed consent.

Answer: We integrated section 2.2 with:

“The registry of sick requests was consulted and, in case of absence from work of the HCWs, a phone call was made to evaluate the need to carry out the swab in relation to the symptoms. as defined by the regional protocol”,

The References section:

“It can be seen that the bibliographic references 1, 3, 4, 6, 9, 12, 13, 18-20, are poorly formulated.

Among the most frequent failures are the missing data in the references of web pages consulted, wrongly put the month and day of publication after the year, the name of the magazines abbreviated with points, doi (in all). For this reason, I encourage you to address.”

Answer: All bibliographic references have been revised according to the journal's specific reference standards

Reviewer 2 Report

The manuscript presents an integrated strategy for the prevention of SARS-CoV-2 infection in healthcare workers. The manuscript is a valuable addition to the scientific literature. The authors may need to present their data in a clear tabular format.

Author Response

Dear Reviewer, thank you for your appreciation of our manuscript.

The manuscript presents an integrated strategy for the prevention of SARS-CoV-2 infection in healthcare workers. The manuscript is a valuable addition to the scientific literature. The authors may need to present their data in a clear tabular format.

Answer: We have improved the legends of the tables and figures and followed the journal’s specific guidelines. We believe that others tables/figures may burden the paper.

Thank you

Anna Maria Cattelan 

Reviewer 3 Report

Cattelan et al. present a timely report on the observations from the Padua University Hospital and how the proper measures of protection of the HCWs and separation of potential Covid patients, helped to prevent further spreading of infections. The presentation of the observations is adequate and the conclusions can lead to replication of the measures in other locations to help in the control of the virus transmission around the world. Nevertheless, I would like to point a couple of minor opportunity areas that should be considered by the authors:

-Include the name of the hospital in the abstract and the introduction, as it is until the Materials and Methods section that the scope of the sample is stated. In the current form, it could be misleading and imply the study covered the whole country.

-Line 132, correct "In addiction" to "In addition"

-Lines 145-151, if the used probes/primers for the RT-PCR are from one of the available kits, provide the reference. If the primers were designed in-house, provide the sequences of such primers. The purpose of this information is to contextualize the possible false positive and false negative rates that could have affected the study.

-Figure 3, edit the vertical axes titles to more meaningful legends, for instance "% Positive tests" and "# performed swabs"

-Line 209-211, does the number '122 person/hours' mean '122 people/hour'? I mean, does the sentence mean the team in the shift worked to attend 122 people per hour? Otherwise, could this sentence be rewritten to convey better the message?

Author Response

Dear Reviewer,

thank you for your appreciation of our manuscript. We have accepted your good suggestions.

- “Include the name of the hospital in the abstract and the introduction, as it is until the Materials and Methods section that the scope of the sample is stated. In the current form, it could be misleading and imply the study covered the whole country”

Answer: The abstract and the introduction were modified accordingly.

- Correct "In addiction" to "In addition".

Answer: done

- “Lines 145-151, if the used probes/primers for the RT-PCR are from one of the available kits, provide the reference. If the primers were designed in-house, provide the sequences of such primers. The purpose of this information is to contextualize the possible false positive and false negative rates that could have affected the study”

Answer: we Integrated the period as follows:

All nasopharyngeal swab samples were processed with an in-house real-time RT-PCR method according to Lavezzo et al (Lavezzo, E., Franchin, E., Ciavarella, C. et al. Suppression of a SARS-CoV-2 outbreak in the Italian municipality of Vo’. Nature (2020). https://doi.org/10.1038/s41586-020-2488-1).

- “Figure 3, edit the vertical axes titles to more meaningful legends, for instance "% Positive tests" and "# performed swabs"

Answer: done

- “Line 209-211, does the number '122 person/hours' mean '122 people/hour'? I mean, does the sentence mean the team in the shift worked to attend 122 people per hour? Otherwise, could this sentence be rewritten to convey better the message?”

Answer: The mistake was corrected, is referred to the sum of hours worked by the staff a day.

Yours sincerely

Anna Maria Cattelan

Reviewer 4 Report

Comments and suggestions for authors

This study investigated the occurrence of SARS-COV-2 infection among HCWs involved in the management of infected patients and the measures adopted to prevent the transmission in the Padua University Hospital, Italy. The study involved 7595 patients and the HCWs team composed of 60 members. Background, research design, methods, results and conclusion are appropriate, adequately described, and clearly presented.

Only few main comments:

  1. In case of a symptomatic subject with a swab negative which decision was taken?
  2. There is no data on ventilation of the “Advanced Triage of the Infectious Diseases Unit”. Taking in account that the suspected SARS-COV-2 cases and contacts wait the result of the test in the dedicated area for an average of 6-8 hours, with a median time of exposure of HCWs to potential risk estimated to be around 8-10 hours, the inhalation of aerosols can´t be excluded, outside aerosol-generating procedures. For the indoor spaces, open windows is considered a efficacy way to avoid the risk of SARS-COV-2 infection transmission by aerosols.
  3. All HCWs nasopharyngeal swabs were negative, and for none of them were reported clinical or virological evidences of infection. Whereas more or less 35% of SARS-COV-2 infections is asymptomatic and the sensitivy of nasopharyngeal swabs has wide variability, why not the inclusion of a serologic test to HCWs?

Only two minor comments:

Line 34 – correct to SARS-COV-2

Table 2 – correct to Total

References

Lines 364, 379, 391 and 401 – do not use capital letters (first letter of the words) to write the tittles of the papers.

Lines 401 and 409 – reduce from 7 to 6 the number of authors.

Author Response

Dear Reviewer,

thank you for your appreciation of our manuscript.

We have accepted your good suggestions.

  1. In case of a symptomatic subject with a swab negative which decision was taken?

Answer: the assessment was made case by case. If the suspicion of COVID-19 was high, other diagnostic tests, such as radiological investigations and the search for SARS-COV-2 in the bronchoalveolar lavage or tracheal aspirate were performed. According to the clinical conditions. the patients were then hospitalized to a General Medicine ward, preferably in a single room pending further diagnostic tests.

  1. There is no data on ventilation of the “Advanced Triage of the Infectious Diseases Unit”. Taking in account that the suspected SARS-COV-2 cases and contacts wait the result of the test in the dedicated area for an average of 6-8 hours, with a median time of exposure of HCWs to potential risk estimated to be around 8-10 hours, the inhalation of aerosols can´t be excluded, outside aerosol-generating procedures. For the indoor spaces, open windows is considered a efficacy way to avoid the risk of SARS-COV-2 infection transmission by aerosols.

Answer: We introduced the following period in the discussion section:

In our tent structures an adequate ventilation was guaranteed by open windows, as it is possible to see in Figure 1.

  1. All HCWs nasopharyngeal swabs were negative, and for none of them were reported clinical or virological evidences of infection. Whereas more or less 35% of SARS-COV-2 infections is asymptomatic and the sensitivy of nasopharyngeal swabs has wide variability, why not the inclusion of a serologic test to HCWs?

Answer: This is a good point. Unfortunately, during this early epidemic phase, validated serologic tests were not available.

Only two minor comments:

Line 34 – correct to SARS-COV-2

Answer: Done

Table 2 – correct to Total

Answer: Done

References

Lines 364, 379, 391 and 401 – do not use capital letters (first letter of the words) to write the tittles of the papers.

Lines 401 and 409 – reduce from 7 to 6 the number of authors.

Answer: All the references were modified accordingly

Yours sincerely

Anna Maria Cattelan

Reviewer 5 Report

The data presented are very interesting and highly topical and useful for defining and managing surveillance sistems on personnel (HCWs) exposed to COVID 19 risks.

The only suggestion concerns the opportunity to complete point 2.3 statistical analyzes with the bibliographic indications

of the formulas and data management used; the indication of a commercial software is not sufficient to assert the numerical results

Author Response

Dear Reviewer,

thank you for your appreciation of our manuscript.

The only suggestion concerns the opportunity to complete point 2.3 statistical analyzes with the bibliographic indications of the formulas and data management used; the indication of a commercial software is not sufficient to assert the numerical results”

Answer: The part of statistical analysis has been changed as follows:

“Data were presented as percentages for categorical variables and mean ± SD (Standard Deviation) for continuous variables, which were compared using Student’s t test for unpaired data, performing a priori tests for equality of variances. A linear regression model was used to analyze the correlation between the number of positive nasopharyngeal swabs and the number of tests performed to HCWs. A p value <0.05 was accepted as statistically significant”.

Yours sincerely

Anna Maria Cattelan

Reviewer 6 Report

This is a very good paper and I recommend it for acceptance. However I have a few little changes. 

1. In a couple places the authors use the term "important number". I would change this to actually giving an estimate of the number, ofr maybe using a different term such as "substantial" or "large" instead of "important".  2. Figure 2 is poorly explained in the caption and in the text. Usually when one speaks of "showing a distribution", a histogram is used. But this is apparently showing, for each age group, the percentage of people in that age group who tested positive? Is this more informative than a histogram of the ages of the people testing positive? As it is now, I'm not sure what Figure 2 really tells us.  3. On p7, in Fig4, is each point representing a day? Maybe this could be clarified in the caption. Also, in the text, why do you say it "did not show a linear association"? Do you mean the relationship is nonlinear, or that the association is not statistically significant? I suspect the latter, but then you should report the p-value associated with the correlation of √.0251.  4. The main result here is the lack of positive cases among the HCW in this study, but the statistical analysis does not focus on this at all. I think the results are impressive, but somewhere in the results you should probably compute a p-value, which will be impressively low I think, showing the probability of 0 positive cases aong your HCW if the overall rate were really 6% as in the Netherlands or 9% as in the rest of Italy. 

Author Response

Dear Reviewer,

thank you for your appreciation of our manuscript. We accepted your suggestions.

  1. In a couple places the authors use the term "important number". I would change this to actually giving an estimate of the number, for maybe using a different term such as "substantial" or "large" instead of "important". 

Answer: done

  1. Figure 2 is poorly explained in the caption and in the text. Usually when one speaks of "showing a distribution", a histogram is used. But this is apparently showing, for each age group, the percentage of people in that age group who tested positive? Is this more informative than a histogram of the ages of the people testing positive? As it is now, I'm not sure what Figure 2 really tells us. 

Answer: We included in the figure the absolute number of patients tested for SARS-CoV-2 and adjusted the period as follows:

“The number of subjects tested for SARS-CoV-2 and the percentage of positive patients attending the Advance Triage Unit according to age group is shown in Figure 2”.

  1. On p7, in Fig4, is each point representing a day? Maybe this could be clarified in the caption. Also, in the text, why do you say it "did not show a linear association"? Do you mean the relationship is nonlinear, or that the association is not statistically significant? I suspect the latter, but then you should report the p-value associated with the correlation of √.0251. 

Answer: We have better explain in the text:

“The regression analysis did not show a statistically significant association between the rate of positivity and the number of nasopharyngeal swabs performed to HCWs (R2=0.025; p=0.27)”.

We added in the cap of Figure 4

Each point represents a day of the period study.

4.The main result here is the lack of positive cases among the HCW in this study, but the statistical analysis does not focus on this at all. I think the results are impressive, but somewhere in the results you should probably compute a p-value, which will be impressively low I think, showing the probability of 0 positive cases among your HCW if the overall rate were really 6% as in the Netherlands or 9% as in the rest of Italy. 

Answer: It is not a comparative study. We can compare our result with those reported in the literature like we described in the discussion. We have highlighted this impressive result introduced the following period in the result section:

“All the swabs performed to HCWs resulted negative for SARS-CoV-2 and none of the HCWs reported symptoms of infection showing a “zero” SARS-COV-2 transmission rate among the HCWs team”.

Yours sincerely

Anna Maria Cattelan

Round 2

Reviewer 1 Report

Dear authors, after reviewing the manuscript, I submit the following comments.

Best regards,

The References section:

It can be seen that the bibliographic references 5, 10, 19, are poorly formulated. According to the MDPI Reference List and Citations Style Guide (URL https://mdpi-res.com/data/mdpi_references_guide_v5.pdf), no publication day and month should be entered. I suggest that you review this and all other references. I give you the structure of the guide in the style of references in Journal Articles.

Author 1; Author 2; Author 3; etc. Title of the article. Journal Abbreviation Year, Volume, Firstpage – Lastpage, doi: prefix / suffix.

Author Response

Dear Reviewer,

Thank you for your additional suggestions.

We adjusted all the references according to the “MDPI Reference List and Citations Style guidelines”. Unfortunately, for references N°10  and N° 19, volume and pages are not yet available because there are still online versions.

5. Wu, Z.; McGoogan, J.M. Characteristics of and Important Lessons From the Coronavirus Disease 2019 (COVID- 19) Outbreak in China: summary of a report of 72 314 cases from the Chinese Center for Disease Control and Prevention. JAMA 2020, 323, 1239-1242, doi:10.1001/jama.2020.2648.

10. Ran, L.; Chen, X.; Wang, Y.; Wu, W.; Zhang, L.; Tan, X. Risk factors of healthcare workers with Corona Virus Disease 2019: A Retrospective cohort study in a designated hospital of Wuhan in China. Clin Infect Dis 2020, doi:10.1093/cid/ciaa287. (Published online ahead of print).

19.Bahl, P.; Doolan, C.; de Silva, C.; Chughtai, A.A.; Bourouiba, L.; MacIntyre, C.R. Airborne or droplet precautions for health workers treating COVID-19? J Infect Dis 2020, doi: 10.1093/infdis/jiaa189. (Published online ahead of print).

Best regards

Anna Maria Cattelan